# Assessment of the Knowledge and Approach of General Dentists Who Treat Children and Pediatric Dentists Regarding the Proper Use of Antibiotics for Children

**DOI:** 10.3390/antibiotics10101181

**Published:** 2021-09-28

**Authors:** Moran Rubanenko, Sigalit Blumer, Kayan Maalof, Shlomo Elbahary, Lazar Katz, Johnny Kharouba

**Affiliations:** Department of Pediatric Dentistry, Faculty of Medicine, School of Dentistry, Tel Aviv University, Tel Aviv 6997801, Israel; moran42@gmail.com (M.R.); blumer@012.net.il (S.B.); kayanmaalouf@mail.tau.ac.il (K.M.); shlomodent@gmail.com (S.E.); lazarkat@tauex.tau.ac.il (L.K.)

**Keywords:** antibiotics, dentists, over-prescribing, antimicrobial resistance, recommended practice

## Abstract

**Background:** Antibiotics are widely used in dentistry. Dentists often provide antibiotics unnecessarily. Excessive use can induce resistant bacterial strains. There are certain indications for the proper use of antibiotics for pediatric dentistry according to the European and American Pediatric Dentistry (EAPD and AAPD). Very often dentists do not follow these guidelines. **Objectives:** This study aims to examine the level of knowledge among general dentists (who also treat children) and pediatric dentists on proper use of antibiotics. In addition, we examined whether there is unjustified use of antibiotics, if dentists are aware of the new and conservative approach of administering antibiotics to patients, and whether there is a relationship between years of professional seniority and dentist’s knowledge level of proper use of antibiotics. **Methods:** One hundred general dentists (GD) who treat children in addition to 100 pediatric dentists (PD) completed the study questionnaires which measured knowledge, practice and attitudes regrading using antibiotics during dental treatment among children. **Results:** The general average of level of knowledge on proper use of antibiotics among general dentists was relatively low for both GD (60.7%) and PDs (65%). PDs demonstrated a relatively greater knowledge of correct use of antibiotics, especially in cases of endodontics and trauma. PDs also showed higher awareness to latest guidelines for the admission of “prophylactic antibiotics” according to the American Association (AAPD) and/or the European Union (EAPD) compared with GDs (86.2% vs. 66.3%). **Conclusion:** The level of knowledge of both general dentists and pediatric dentists is poor, with a large percentage of dentists from both groups not knowing whether antibiotics are needed in a specific dental case or not. Compliance with the EAPD/AAPD guidelines is also low and inadequate. The method of prescribing antibiotics given by dentists can be improved by increasing awareness, educational initiatives, and postgraduate courses among dentists regarding the recommended indications.

## 1. Introduction

Antibiotics are widely used in medicine and dentistry for both preventive and therapeutic purposes. Excessive use can induce resistant bacterial strains. Increasing bacterial resistance to antibiotics is well documented in children and poses a serious concern in global health [1,2]. The National Center for Disease Control and Prevention estimate that approximately one-third of all outpatient antibiotic prescriptions are unnecessary [3]. Antibiotic overuse might be associated with increased risk of allergy-related disease [4]. Antibiotic underuse may also lead to prolonged disease duration and increased rate of disease-related complications, both of which may be avoided with prompt treatment of the bacterial infection [5]. In the United States, dentists prescribe about 10% of prescriptions of all common antibiotics, which is considered to be the fourth highest prescriber of antibiotics in the United States by volume [6,7]. Hence, it is important that dentists have sufficient knowledge and awareness of the use of antibiotics.

In order to make it easier for physicians in the various medical fields, medical and dental associations issue and update guidelines that direct practitioners to decide in which conditions it is recommended to use antibiotics [8]. Pediatric dentists in many countries around the world are assisted by guidelines regarding the proper administration of antibiotics by the American Association and the European Association of Pediatric Dentists [9]. 

Data from different countries has shown that there is a large variance in dentists’ knowledge of clinical conditions that require taking antibiotics [10,11,12,13]. For example, in a survey conducted by Oberoi [14], results showed that 39% of dentists and 27% of physicians strictly followed antibiotic treatment guidelines. Many physicians rely on the recommendation of other physicians or decide that when in doubt give patients the most conservative dose [14]. The method of prescribing antibiotics given by dentists can be improved by increasing the awareness among dentists about the recommended indications. [4]. To the best of our knowledge, there is no information about the use of antibiotics by dentists and their adherence to dental organizations guidelines.

The primary end point of this study is to assess the knowledge, practice and attitudes regrading using antibiotics during dental treatment among children. Specifically, we aim to compare between general dentists who treat children and dentists specialized in dentistry of children about antibiotics after dental treatment. The secondary end point was to check if there is an unjustified use of antibiotics by pediatric dentists according to the guidelines of AAPD and EAPD.

## 2. Materials and Methods

### 2.1. Participants and Procedure 

To calculate the sample size required for this stud, we used G-power software with the following assumptions: type 1 error of 5%, desired power of 80% and a moderate effect size for the difference in knowledge between general dentists and pediatric dentists (Cohen’s d = 0.35). The desired sample under these assumptions were 204 participants. 

Two hundred dentists from Israel participated in this study out of 225 questionnaires that were distributed (indicating 88% response rate). Half of the sample (N = 100) were 100 general dentists (GD) who treat children, and the other half (N = 100) were pediatric dentists (PD). 

This study was approved by the Ethical Committee of Tel Aviv University. Following the ethical approval, participants completed the study’s questionnaires at the national conference of the Israel association of dentistry and the annual conference of the Israel society of pediatric dentistry and also by distributing them for instructors at the dental school of Tel Aviv university.

### 2.2. Measures 

Data gathered using a questionnaire that was constructed based on previous questionnaires [14,15]. The questionnaire included the following parts:

Demographic information: Sex, age, years of experience, main population of treatment (children vs. adults), type of clinic (public, private, academic).

General Knowledge regarding antibiotics’ prescribing: Dentists were asked in which clinical conditions it is recommended to use antibiotics following a dental treatment (endodontics, trauma, displacements and systemic diseases). In addition, dentists were asked about their awareness regarding the instructions of prescribing antibiotics according to the guidelines and updates of AAPD and/or EAPD.

Practices of using antibiotics. Dentists were asked to in which clinical conditions they actually use antibiotics following dental treatments (for example pulpitis, localized intraoral swelling). 

### 2.3. Statistical Analysis 

Data were analyzed using SPSS software versions 27. Descriptive statistics were used to describe the percentage distribution of the answers to the various questions in the questionnaire on the correct use of antibiotics. Differences between the groups (GD vs. PD) were calculated using independent t-tests (for continuous variables), and chi-square tests (for categorical variables); *p*-values lower than 5% is considered significant. 

## 3. Results

### 3.1. Demographic and Professional Characteristics

Table 1 presents the demographic and professional information about the total sample, and also differences between groups. Groups were similar across most of the variables. 

As expected, the results show that the higher frequency of the PDs (87.0%) treated only children as opposed to treating children and adults (13.0%), in comparison to the GDs who most of them treated children and adults (71.6%) as opposed to treating only children (28.4%) (X^2^ (1) = 64.47, *p* < 0.01). Groups were also found to differ in clinic type (X^2^ (3) = 29.65, *p* < 0.01), while a higher rate of PDs (28.0%) worked both in clinic and academia, in comparison to the GD (3.0%).

### 3.2. Comparison in Knowledge between Groups

Table 2 presents differences in knowledge regarding the use of antibiotics between groups. The results show that PDs (M = 20.18, SD = 4.29) responded correctly on more questions about the uses of antibiotics in comparison to GDs (M = 18.83, SD =3.76), *t* (197) = 2.36, *p* < 0.01). Probing the specific topics showed that PDs demonstrated more knowledge regarding the use of antibiotics at endodontics, trauma and extractions. 

### 3.3. Comparison in Using Guidelines between Groups

Table 3 presents differences in using guidelines of using antibiotics. 

Most of the dentists in the sample (about 70%) reported they are aware to guidelines for prescribing antibiotics. However, high rates of PDs reported they use AAPD and EAPD guidelines for prophylactic antibiotics in comparison with GDs (86.2% vs. 66.3%, *p* < 0.01). 

Higher rate of GDs reported they advise patients to adhere dosage regimen in comparison with PDs (93.9% vs. 81.5%, *p* < 0.01). In addition, in the case of orofacial infection with the administration of antibiotics without improvement in the condition, more GD only drained (57.73%) while more PD drained and gave antibiotics (51.13%), (Χ^2^ (3) = 8.53, *p* = 0.01). 

### 3.4. Comparison in Practices of Using Antibiotics between Groups

Table 4 and Table 5 present significant and non-significant comparisons between groups is the practices of using antibiotics, respectively. Results showed that, in general PDs demonstrate less frequent use of antibiotics in comparison with GDs. Specifically, PD use less antibiotics in the following cases: intraoral sinus tract (16.2% vs. 45.8%), luxation (14.3% vs. 37.5%), extraction by open method (50.6% vs. 79.8%), periapical localized abscess (56.3% vs. 83.5%), apical periodontitis (35.2% vs. 61.4%), dry socket (20.7% vs. 51.1%), duration of antibiotics course (56.8% vs. 28.3%), periodontal disease (45.8% vs. 60.6%), respiratory disorders (22.5% vs. 43.0%), and viral infection (5.5% vs. 13.7%). 

However, higher frequency of use in antibiotics by PDs was found in clinical indications of juvenile diabetes (20.9% vs. 10.0%), congenital cardiac abnormalities (65.8% vs. 51.2%) and avulsion (79.4% vs. 67.0%). 

Similar use in antibiotics were found in the following clinical indications: reversible pulpitis, irreversible pulpitis, simple extraction, cellulitis, localized intraoral swelling, extensive hematoma, extraoral sinus tract, bone fracture, intrusion, pericoronitis limited to tooth, flagyl and amoxicillin/augmantin, blood dyscrasias and subacute bacterial endocarditis.

To assess differences between groups after adjusting for potential biases, we conducted multivariate logistic regression model. The independent variables were all the factors that yielded significant differences (as elaborated in Table 3): intraoral sinus tract, luxation, extraction by open method, periapical localized abscess, apical periodontitis, dry socket, duration of antibiotics course, periodontal disease, juvenile diabetes, respiratory disorders, congenital cardiac abnormalities, avulsion and viral infections. In addition, we controlled for age, gender and years of experience (see Table 6).

Results showed that after adjusting for demographic and professional differences between groups, PD use less in intraoral sinus tract (OR = 0.65), periapical localized abscess (OR = 0.72), apical periodontitis (OR = 0.54), and dry socket (OR = 0.62). PD had longer antibiotics course (OR = 1.88).

## 4. Discussion

This study examined differences in knowledge, attitudes and practices of general dentists (GDs) and dentists specialized in pediatric dentistry (PDs) for proper use of antibiotics.

Overall, PDs demonstrated greater knowledge in comparison with general practitioners in the in regard to the clinical cases that justify the use of antibiotics following a dental treatment. Specifically, PDs demonstrated greater understanding in regard to the domains of endodontics and trauma, compared with GDs. Despite these differences, the overall knowledge about the correct use of antibiotics for the treatment of various clinical conditions in the dental clinic in both groups is insufficient (PDs 65%, GDs 60.7%). In addition, we found that the awareness of the guidelines and updates to guidelines, is mediocre (73% for GDs compared to 75.3% for PDs). Hence, it seems that dentists in both groups think they are more aware than their actual answers within the questionnaire. According to the reports of the participating physicians the level of commitment that complies with the provisions and guidelines of the associations (AAPD/EAPD) is inconsistent, and sometimes without a clear trend. So, in some cases a relatively good standing commitment be in the case of Cellulitis (85% of GDs give antibiotics and 91% of PDs). This figure is consistent with a previous study by Sivabalan [10], which found that 99% of general practitioners and 100% of pedodontics reported they are recommending and giving antibiotics in the case of cellulite. Facial cellulite even if not related to any disorder is a serious disease that needs to be treated with antibiotics immediately because of the possibility of the infection spreading through the lymphatic system and blood circulation, and there is a risk of sepsis. 

When asked about giving antibiotics in reversible pulpitis, 13.4% GDs and 8% of PDs answered that they give antibiotics. This is in contrast to the AAPD/EAPD guidelines. In a study conducted by Sugata, it was found that only 2% of pedodontics reported using antibiotics in the case of reversible pulpitis. It should be noted, however, that in this study the number of dentists who answered this question was low. In the case of irreversible pulpitis, about 20% of GDs and 24.48% of PDs answered that they prescribe antibiotics, which is contrary to AAPD/EAPD guidelines. In the same study by Sugata, 32% of dentists also answered that they give antibiotics in the case of irreversible pulpitis γ.

In another study [11], 45% of dentists give antibiotics in the case of irreversible pulpitis. While in this study there is a lack of knowledge about not giving antibiotics in the case of irreversible pulpitis, but the knowledge is better than previous studies. In cases of intraoral sinus tract, periapical localized abscess, and apical periodontitis, unjustified use of antibiotics is considered. These cases do not require the use of antibiotics but a “surgical” action such as pus drainage or root canal treatment. In addition, there is a statistically significant difference between the two groups, so that in all of them GDs gave more antibiotics than PDs, this can be explained by the additional education that pediatric dentists receive. However, overuse of antibiotics can also be seen in the above cases also among PDs (Intraoral sinus tract 16%, Periapical localized abscess 56%, Apical periodontitis 35%). This indicates a lack of sufficient knowledge in EAPD/AAPD guidelines in both groups. In a study conducted by Sivaraman [11] 68% of participating dentists prescribed antibiotics in the case of local dental abscess with gingival swelling, and in the same survey comparing pedodontics to endodontics in the case of draining fistula found that 39% of pedodontics give prescription compared to only 12% of endodontics. This is attributed in their opinion to the wider knowledge and occupation of endodontics in connection with root canal diseases [11]. 

In this study, we examined the use of antibiotics in five cases of dental trauma: avulsion, intrusion, lateral luxation, extensive hematoma and fracture of the jawbone. The results of the study are inconsistent and there is no clear trend of overuse or lack of use in these cases. However, the level of knowledge regarding the administration of antibiotics in cases of trauma is not good. In both cases of lateral luxation and intrusion there is an overuse of antibiotics in both groups (In intrusion 39% of PDs and 46% of GDs give antibiotics; in lateral luxation 14% of PDs and 37.75% of GDs give antibiotics). 

According to the AAPD/EAPD guidelines no antibiotics are needed in these cases. In both cases, the percentage of GDs giving antibiotics is higher than PDs, but with lateral luxation has a significant difference. This difference can be explained by the fact that pedodontics are more common in these cases so that tooth displacement injuries (such as luxation injuries) are more common in deciduous teeth. In other cases of dental trauma (avulsion, extensive hematoma and jaw fracture) there is a trend of antibiotic deficiency, so a large percentage of dentists answered that they do not give antibiotics while according to AAPD guidelines antibiotics should be given. 

As to the period of administration of the antibiotic to the patient a significant difference was found, so of PDs give antibiotic prescriptions for a longer period of time. The results of the of PDs come by agreement with other study, so that most of the 57% answered that they register a 7-day prescription, and 30% register a 10-day prescription. This is despite the fact that short periods are preferable to particularly long periods when caring for children, as children’s response to long courses is not good. On the other hand, there is a misconception about the use of antibiotics that should be used for a certain number of days to ‘kill the resistant strains’, the vast majority of strains acquire resistance when using under-treated doses or for a very long time. Longer durations—up to 21 days, may result in the survival of resistant strains and a decrease in the ability of the natural oral flora to resist the settlement of harmful microorganisms that are not normal inhabitants and eventually even lead to infections by bacteria and yeast [9].

In cases of viral infection, dentists showed a lack of compliance with the guidelines. Although most agree that antibiotics should not be given in the case of viral infection, still 32.63% of GDs and 5.49% of PDs thought that should be given. According to AAPD guidelines, viral infections such as primary herpetic gingivostomatitis should not be treated with antibiotics. Dentists were asked about two cases, non-congenital heart disease and Subacute bacterial endocarditis that requires prophylactic antibiotics. In the first case, which does not require prophylactic antibiotics before treatment according to AAPD/EAPD guidelines most dentists from both groups do give prophylactic antibiotics. In the case of subacute endocarditis, there is a trend to give antibiotics (69.23% from GD and 79.76% from PD) but it is still considered not sufficient knowledge because this case requires prophylactic antibiotics, and the results show a lack of use [9]. 

In conclusion, this study supports previous studies that lack the level of knowledge of dentists (whether general dentists or pedodontics) about the proper use of antibiotics. There is a need to provide clear guidelines for when and how to use antibiotics, the duration of the period and the right dosage through lectures, courses, or training program and educational initiatives.

## Figures and Tables

**Table 1 antibiotics-10-01181-t001:** Demographic and clinical characteristics of the sample.

	Total Sample		GD		PD		Χ^2^	*p*
	N	%	N	%	N	%		
Age							2.26	0.32
25–30	14	7.1	7	7.2	7	7.1		
31–40	83	42.3	36	37.1	47	47.5		
40+	99	50.0	54	55.7	45	45.5		
Gender							3.83	0.05
Male	75	37.7	44	44.4	31	31.0		
Female	124	62.3	45	55.6	69	69.0		
Years of Experience							7.64	0.05
0–3	15	7.5	10	10.0	5	5.0		
3–6	25	12.5	10	10.0	15	15.0		
7–10	38	19.0	13	13.0	25	25.0		
11+	122	61.0	67	67.0	55	55.0		
Population of Treatment							64.47	<0.01
Children	110	55.0	23	23.0	87	87.0		
Children and adults	71	35.5	58	58.0	13	13.0		
Adults	19	9.5	19	19.0	-	-		
Clinic Type							29.65	<0.01
Public	49	24.5	35	35.0	14	14.0		
Public and private	67	33.5	33	33.0	34	34.0		
Private	53	26.5	29	29.0	24	24.0		
Academic + clinic	31	15.5	3	3.0	28	28.0		
Antibiotics Prescription Last Month							2.17	0.54
Until 5	141	70.5	66	66.0	75	75.0		
6–10	42	21.0	25	25.0	17	17.0		
10–20	13	6.5	7	7.0	6	7.0		
20+	4	2.0	2	2.0	2	2.0		

**Table 2 antibiotics-10-01181-t002:** Differences between PDs and GDs in knowledge of prescribing antibiotics.

	GD	PD	*t*	*p*
	M ± SD	M ± SD		
General knowledge	18.83 ± 3.76	20.18 ± 4.29	2.36	<0.01
Endodontics	2.59 ± 1.13	3.18 ± 1.26	3.26	<0.01
Trauma	2.76 ± 1.29	3.35 ± 1.09	3.75	<0.01
Extractions	3.14 ± 1.25	3.46 ± 1.42	1.69	0.04
Systemic diseases	3.76 ± 1.39	3.75 ± 1.65	0.05	0.97
Awareness to instructions	2.19 ± 0.87	2.26 ± 0.90	0.56	0.29

**Table 3 antibiotics-10-01181-t003:** Differences between the GD and PD at the guidelines.

	GD N (%)	PD N (%)	Χ^2^	*p*
Awareness to guidelines	67 (69.8)	63 (69.2)	0.01	0.93
AAPD and EAPD Guidelines for prophylactic antibiotics	63 (66.3)	75 (86.2)	9.8	<0.01
Advising patients to adhere dosage regimen	92 (93.9)	75 (81.5)	6.81	<0.01
Orofacial infection without improvement			8.53	0.01
Give more antibiotics	5 (5.15)	10 (0.11)		
Drains and antibiotics	35 (36.08)	45 (51.13)		
Drains	56 (57.73)	33 (37.5)		

**Table 4 antibiotics-10-01181-t004:** Significant differences between groups at antibiotics’ prescription by clinical indications.

	GD	PD	Χ^2^	*p*
	N (%)	N (%)		
Intraoral sinus tract	44 (45.8)	16 16.2))	20.14	<0.01
Luxation	37 (37.5)	14 (14.28)	14.02	<0.01
Extraction by Open Method	75 (79.8)	43 (50.6)	16.94	<0.01
Periapical localized abscess	81 (83.5)	53 (56.36)	16.77	<0.01
Apical Periodontitis	59 (61.45)	31 (35.22)	12.64	<0.01
Dry Socket	51 (51.1)	18 (20.68)	17.81	<0.01
Duration of Antibiotics Course	28 (28.28)	54 (56.84)	16.20	<0.01
Periodontal Disease	60 (60.6)	44 (45.83)	4.27	0.04
Juvenile Diabetes	9 (10.0)	18 (20.93)	4.04	0.04
Respiratory Disorders	40 (43.01)	22 (22.58)	5.99	0.01
Congenital Cardiac Abnormalities	44 (51.2)	50 (65.8)	3.54	0.06
Avulsion	65 (67.0)	77 (79.4)	3.78	0.05
Viral infections	13 (13.7)	5 (5.5)	3.57	0.06

**Table 5 antibiotics-10-01181-t005:** Non-significant differences between groups at antibiotics’ prescription by clinical indications.

	GD	PD	Χ^2^	*p*
	N (%)	N (%)		
Reversible pulpitis	13 (13.4)	8 (8)	2.14	0.14
Irreversible pulpitis	19 (19.79)	24 (24.48)	0.62	0.43
Simple Extraction	5 (5.05)	5 (5.2)	0.00	0.96
Cellulitis	85 (87.62)	91 (92.85)	1.51	0.22
Localized Intraoral Swelling	50 (52.6)	47 (53)	0.42	0.52
Extensive Hematoma	46 (46.5)	56 (57.7)	2.49	0.11
Extraoral sinus tract	63 (64.9)	56 (58.3)	0.89	0.34
Bone Fracture	52 (53.1)	55 (56.1)	0.18	0.66
Intrusion	45 (46.4)	39 (39.4)	0.98	0.32
Pericoronitis limited to tooth	54 (55.1)	53 (55.2)	0.03	0.87
Flagyl and Amoxicillin/Augmantin			2.78	0.25
Dalacin	0	5 (11.62)		
Flagyl	20 (55.55)	18 (41.86)		
Amoxicillin/Augmantin	16 (44.44)	20 (46.51)		
Duration of Prescription of Anaerobic Infection	21 (50)	26 (59)	0.72	0.40
Blood Dyscrasias	36 (39.65)	26 (30.23)	1.69	0.19
Subacute Bacterial Endocarditis	63 (69.2)	67 (79.8)	2.54	0.11

**Table 6 antibiotics-10-01181-t006:** Odds ratio for predicting group classification (PD = 1, vs. GD = 0).

		95% CI		
	OR	LL	UL	*p*
Age (31+)	1.03	0.89	1.18	0.49
Gender (F)	1.10	0.92	1.29	0.67
Years of experience (7+)	1.22	1.08	1.78	0.03
Intraoral sinus tract	0.65	0.31	0.87	0.02
Luxation	0.85	0.66	1.31	0.24
Extraction by Open Method	0.89	0.78	1.21	0.57
Periapical localized abscess	0.72	0.59	0.95	0.01
Apical Periodontitis	0.54	0.12	0.78	0.01
Dry Socket	0.62	0.48	0.89	0.02
Duration of Antibiotics Course	1.88	1.45	2.55	0.01
Periodontal Disease	0.82	0.35	1.42	0.35
Juvenile Diabetes	0.78	0.55	1.19	0.23
Respiratory Disorders	1.12	0.78	2.66	0.62
Congenital Cardiac Abnormalities	1.12	0.89	1.45	0.37
Avulsion	1.18	0.87	1.55	0.58
Viral infections	0.87	0.45	1.41	0.74

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
