# Peer review of "Assessment of the Knowledge and Approach of General Dentists Who Treat Children and Pediatric Dentists Regarding the Proper Use of Antibiotics for Children"

_antibiotics, 2021, doi:10.3390/antibiotics10101181_

Round 1
Reviewer 1 Report
Dear author
This study aims to examine the level of knowledge among general dentists (who also treat children) and pediatric dentists on proper use of antibiotics. However, there are biases in the sample that make it difficult to generalize. There are also differences in the years of experience and gender ratio for the comparison between PD and GP. These cannot rule out the possibility that these may have affected the results. Therefore, I am convinced that the comparison between PD and GP does not present any useful findings. I think this study has many limitations, but the author does not list them. Unfortunately, I did not find any interesting findings in this article.
Author Response
Response to reviewer 1
Thank you for your useful comments.
Agree that the groups differ in demographic characteristics, and this is one of the limitations of the study. (We added this limitation to the discussion).
On the other hand, the groups of the study were randomly selected and therefore it is not possible to guarantee that the groups will be similar. This is a limitation in any study. Larger groups had to be taken and then maybe the groups would be more similar.
The main objective of the study was to examine the level of knowledge of both groups in the correct use of antibiotics. The important result was that in both groups the level of knowledge was low and needs to be improved. And these are groups of dentists who go to conferences and are active in academia. Therefore, we expect that the knowledge of a general dentist who is not updated is even lower. (We added this comment to the discussion)

Reviewer 2 Report
The manuscript needs an in-depth revision. The methods section needs to be revised in order to understand what was actually analyzed. Inclusion criteria are not defined and it is not explained how knowledge, practices, and attitudes were assessed. It is only stated that a questionnaire was applied and that "Dentists were asked in which clinical conditions it is recommended to use antibiotics following a dental treatment (Endodontics, Trauma, Extractions, and Systemic diseases). In addition, dentists were asked about their awareness regarding the instructions of prescribing antibiotics according to the guidelines and updates of AAPD and/or EAPD." How was assessed the awareness? Questions "yes" or "no"? What is the cut-off for having or not having knowledge?
Authors also refer that: "Practices of using antibiotics. Dentists were asked to in which clinical conditions they use antibiotics following dental treatments (for example pulpitis, localized intraoral swelling). Why these conditions are different from conditions about knowledge??
And how were attitudes assessed?
Perhaps it would be important in addition to a more detailed description of the methods and clearer presentation of the results, to also put the questionnaire in the supplementary material.
On the other hand, the sample seems to be quite biased since the questionnaires were applied at the national conference of the Israel Association of dentistry and the annual conference of the Israel society of pediatric dentistry and by distributing them for instructors at the dental school of Tel Aviv University. A sample of doctors attending a conference or working in a university might represent the most attentive and concerned professionals. This should be considered as a limitation of the study.
Author Response
Dear reviewer (2)
Thank you for your kind remarks.
Hereby our comments for each of your remarks:
- Inclusion criteria – Half of the sample (N=100) were100 general dentists (GD) who treat children (but does not have a post-graduation degree at pedodontics) , and the other half (N=100) were pediatric dentists (who are at the post-graduation program or finished it) – those who do not treat children were excluded from the study
- The cut off for having or not having knowledge – was assessed by answering correctly to more than 50% of the questions right.
- we agree ,there is overlap between knowledge and awareness but there is difference between knowing the guidelines and implementing them.(knowledge versus practice and attiude).we assumed questions no 10,15,16,20represents knowledge, questions 11,12,14 represents awareness ,questions 9,18,19,21,13represents practice and attitude.
- More detailed method.
Added
The questionnaire included questions aimed at testing physicians' level of knowledge regarding the correct use of antibiotics. If they are aware of the concept of "antibiotic resistance". Or what is the optimal duration of administration of antibiotics in odontogenic infection, or if they give antibiotics in clinical situations such as pulpitis (Reversible pulpitis, Irreversible pulpitis),sinus tract, intraoral swelling, Cellulitis, Dental trauma , Periodontal disease, extractions, Periapical abscess or in systemic situations such as viral infections, Juvenile diabetes, Blood dyscrasias and lung diseases, cardiovascular disease such as Congenital heart disease, Subacute bacterial endocarditis.
The questionnaire also contained questions that examined how in practice physicians would establish their knowledge in the care of children, such as whether they monitor at the time of antibiotic treatment or explain to the patient about the correct dose and time of treatment or what they do if antibiotic treatment does not improve patient condition or if they ask the patient if he has taken antibiotics last week before prescribing antibiotics.
- We were asked to add the questionnaire to the supplementary material
Added
- The questionnaire also distributed to dentists conferences and at Tel Aviv university. Moreover, we assume that the dentists which attends conferences are the most updated. If their knowledge is not sufficing, then we can assume other dentists has even lesser knowledge than theirs.
Added to the discussion
If there's any further remarks, we will be happy to clear them
Reviewer 3 Report
An interesting article evaluating the level of antibiotic prescription for children in general and pediatric dentists is presented.
The authors rightly state that the problem of bacterial resistance to antibiotics is one of the most important problems in today's healthcare. This problem must be solved at all levels and the Antibiotic Stewardship application is a part of the solution. The topic of the article fits into this framework and can be considered as very actual.
However, I have three key comments on the text:
- Rational application of antibiotics is part of a comprehensive approach that forms the Antibiotic Stewardship. I consider it appropriate for this term to be included in the text and, at the same time, for the results to be discussed in this context.
- Table 4 is listed twice, while Table 5 is completely missing.
- In the case of antibiotics (they are incorrectly listed in the evaluation according to clinical indications, see Table 4), it is more appropriate to state the basic names of antibiotics, ie metronidazole and not Flagyl. I do not understand the term Amoxicillin / Augmentin, these are two different antibiotics, namely amoxicillin and amoxicillin / clavulanic acid.
I consider it necessary to revise the manuscript on the basis of the above comments.
Author Response
Response to the three key comments on the text of reviewer 1
1-Thank you for informing me about antibiotic stewardship program. I included in the introduction (second paragraph):
" A growing body of evidence demonstrates that hospital-based programs dedicated to improving antibiotic use, commonly referred to as “Antibiotic Stewardship Programs (ASPs),” can both optimize the treatment of infections and reduce adverse events associated (8). These programs help clinicians improve the quality of patient care and improve patient safety (9). They also significantly reduce hospital rates of CDI (Clostridium difficile infection) and antibiotic resistance.(10)
And in the discussion (last paragraph before conclusions): Core Elements of Hospital Antibiotic Stewardship Programs include Accountability, Drug Expertise, Action, tracking, Reporting, and Education. Leadership Commitment (8).
This program is mostly implemented in hospitals. Dentists usually work in private practices, but they should be acquainted with this substantial program. It will be of great benefit if it could be implemented in dental schools or dental associations.
- Corrected. Table 4 now is only one and there is table 5
3- Corrected. I stated the basic names of antibiotics
Round 2
Reviewer 1 Report
Dear author
Thank you for your consideration of my previous peer review comments. The author has made corrections to the extent possible.
However, I do not think that this article should be published in an international journal. The author stated that the important result was that in both groups the level of knowledge was low and needs to be improved. What is the basis for the author's judgment that the level of dentists in the country is low? First of all, the author should publish the results of this study in a domestic journal to make domestic dentists aware of the problem you have noticed. Authors should consider adjusting for bias between the two groups using multivariate analysis or other methods.
Author Response
Dear reviewer 1
Thank you first of all for your helpful comments. Here are our responses
The author stated that the important result was that in both groups the level of knowledge was low and needs to be improved. What is the basis for the author's judgment that the level of dentists in the country is low?
The level of knowledge was low in both groups because in most questions there was overuse and unjustified prescription of antibiotics as in cases of: reversible pulpitis, irreversible pulpitis, intraoral sinus tract, periapical localized abscess, and apical periodontitis, viral infection and other clinical situations.
We also adopted a questionnaire that was used in previous studies, and this underscores the validity of the questionnaire whose purpose was to test knowledge about antibiotics.
However, I do not think that this article should be published in an international journal.
The benefit of this study might be that other similar studies in different countries will be done. Then we can conclude that there is a problem of misuse of antibiotic worldwide.
Quite a few works on local populations are published in international journals. For example:
Elhennawy K, Anang M, Splieth C, Bekes K, Manton DJ, Hedar Z, Krois J, Jost-Brinkmann PG, Schwendicke F. Knowledge, attitudes, and beliefs regarding molar incisor hypomineralization (MIH) amongst German dental students. Int J Paediatr Dent. 2021 Jul;31(4):486-495. doi: 10.1111/ipd.12715. Epub 2020 Sep 11. PMID: 32813919.
Kind LS, Aartman IHA, van Gemert-Schriks MCM, Bonifacio CC. Parents' satisfaction on dental care of Dutch children with Autism Spectrum Disorder. Eur Arch Paediatr Dent. 2021 Jun;22(3):491-496. doi: 10.1007/s40368-020-00586-y. Epub 2021 Jun 18. PMID: 33382440; PMCID: PMC8213657.
The author should publish the results of this study in a domestic journal to make domestic dentists aware of the problem you have noticed.
We agree with your proposition to publish the results of this study in a domestic journal, but we will do that after it will be accepted in an international journal.
Authors should consider adjusting for bias between the two groups using multivariate analysis or other methods
Regarding a multivariate model, we thought about it originally, but to build a multivariate model one must decide on one central outcome that is predicted. This is a bit difficult in the case of this specific study because we have examined many areas of knowledge.
We added this limitation in the discussion.

Reviewer 2 Report
“Clostridium difficile” should be in italic, please change for “Clostridium difficile”
On page 2 please change “stud” to “study”.
On the other hand, the groups of the study were randomly selected and therefore it is not possible to guarantee that the groups will be similar.
For some diseases, the authors use capital letters, for others do not. Please do not use never.
In the discussion section, the authors write “On the other hand, the groups of the study were randomly selected and therefore it is not possible to guarantee that the groups will be similar.” The sample is a convenience sample and not a randomized sample. Please remove this sentence. And rewrite the following sentence, adding that the sample is not a randomized one and that the result cannot be extrapolated.: “Another limitation is that the questionnaires were applied at the national conference of the Israel Association of dentistry and the annual conference of the society of pediatric dentistry and by distributing to them for instructors at the dental school in Tel Aviv University. This sample of dentists might represent the most attentive and concerned
professionals.”
The authors mention that they have put the questionnaire in supplementary material, but I cannot find it. Please add it.
Author Response
Dear reviewer 2
Thank you for your helpful comments. Here are our responses
1-Clostridium difficile” should be in italic, please change for “Clostridium difficile”
Corrected to “Clostridium difficile”
2-On page 2 please change “stud” to “study”.
corrected
3- For some diseases, the authors use capital letters, for others do not. Please do not use never.
corrected
4-In the discussion section, the authors write “On the other hand, the groups of the study were randomly selected and therefore it is not possible to guarantee that the groups will be similar.” The sample is a convenience sample and not a randomized sample. Please remove this sentence.
removed
And rewrite the following sentence, adding that the sample is not a randomized one and that the result cannot be extrapolated.
We added it as limitation
“Another limitation is that the questionnaires were applied at the national conference of the Israel Association of dentistry and the annual conference of the society of pediatric dentistry and by distributing to them for instructors at the dental school in Tel Aviv University. This sample of dentists might represent the most attentive and concerned professionals
We added to the limitations
The authors mention that they have put the questionnaire in supplementary material, but I cannot find it. Please add it.
We added it.
Reviewer 3 Report
Unfortunately, I cannot recommend accepting the manuscript in its current form.
I have two comments:
- The term CDI (Clostridium difficile infection) is wrong, CDI (Clostridioides difficile infection) is correct
- There are still uncertainties in Table 5. Differences by clinical indications are listed here. What are the clinical indications Dalacin, Metronidazole and the Duration of antibiotic therapy? These are not clinical indications!
Author Response
Response to reviewer
The term CDI (Clostridium difficile infection) is wrong, CDI (Clostridioides difficile infection) is correct
Corrected in the text.
There are still uncertainties in Table 5. Differences by clinical indications are listed here. What are the clinical indications Dalacin, Metronidazole and the Duration of antibiotic therapy? These are not clinical indications
Doctors were asked what antibiotics they would give in case of anaerobic infection. Amoxicillin or Metronidazole or Dalacin.
The duration belongs to knowledge. Deleted from table 5